# Towards Real-World Streaming Speech Translation for Code-Switched Speech

**Belen Alastruey**[1]*, **Matthias Sperber**[2], **Christian Gollan**[2],
**Dominic Telaar**[2], **Tim Ng**[2], **Aashish Agargwal**[2]

[1]TALP Research Center, Universitat Politècnica de Catalunya, Barcelona

[2]Apple

belen.alastruey@upc.edu, sperber@apple.com

## Abstract

Code-switching (CS), i.e. mixing different languages in a single sentence, is a common phenomenon in communication and can be challenging in many Natural Language Processing (NLP) settings. Previous studies on CS speech have shown promising results for end-to-end speech translation (ST), but have been limited to offline scenarios and to translation to one of the languages present in the source (*monolingual transcription*).

In this paper, we focus on two essential yet unexplored areas for real-world CS speech translation: streaming settings, and translation to a third language (i.e., a language not included in the source). To this end, we extend the Fisher and Miami test and validation datasets to include new targets in Spanish and German. Using this data, we train a model for both offline and streaming ST and we establish baseline results for the two settings mentioned earlier.

## 1 Introduction

Speech technologies are one of the main applications of machine learning, and are currently deployed in many real-world scenarios. To ensure a adequate user experience, factors other than accuracy need to be taken into account. One of them is the ability to produce an output in real-time (streaming settings) with a low latency and another one is effectively handling the distinctive characteristics inherent in spoken language, like Code-switching. Code-switching (CS) is the phenomenon in which a speaker alternates between multiple languages in a single utterance. Due to globalization (Winata et al., 2022), it is becoming increasingly prevalent in spoken language, not only in bilingual communities but also in monolingual communities.

CS presents a challenge in various natural language processing (NLP) settings, such as automatic speech recognition (ASR), machine translation (MT), and speech translation (ST), due to

the inherent complexity of dealing with two source languages, as well as the scarcity of CS training and test data (Jose et al., 2020).

Despite the relevance of ST for CS speech task, the available literature on the subject is rather limited. Nakayama et al. (2019) investigate the task defined as *monolingual transcription*, i.e. transcribing a CS utterance using words of only one language, hence translating those words that are CS. Their work proposes and compares different approaches to evaluate the stated task in Japanese-English CS to English. Other follow-up work takes a similar approach (see Section 2).

To date, however, certain essential topics, such as translation to a language not present in the CS source or streaming ST, have yet to be explored, despite its critical importance for real-world usage. The primary challenge in translating to a third language stems from the unavailability of datasets with such characteristics. Furthermore, streaming settings present further challenges: achieving a balance between latency, stability and accuracy is crucial for delivering a seamless user experience, as with any streaming task. Besides, CS tasks may require more context than monolingual ones because of the added complexity of language mixing. Thus, addressing the trade-offs between these metrics in CS streaming ST may prove to be more intricate than with monolingual data.

In our work, we resolve the two aforementioned challenges: first, the insufficiency of data and results for translation to a third language, and second, the absence of a baseline for streaming CS ST.

To alleviate the data scarcity in CS tasks, we extend Fisher (Cieri et al., 2004) and Bangor Miami CS (Deuchar et al., 2014) datasets (combined English and Spanish source and English targets) by incorporating Spanish and German targets in the test and validation sets.[1] These additions allow

---

*Work done during an internship at Apple.

[1]Data available at https://github.com/apple/ml-codeswitching-translations.

us to evaluate the performance of our models on monolingual transcription (translation to English or Spanish), but also for the first time in CS ST into a third language (German) setting baseline results.

Furthermore, this study is the first on streaming ST for CS speech, and examines errors in transcripts generated by both offline and streaming models, considering different latency and flickering constraints, and different training techniques such as prefix-sampling. We show that prefix-sampling does not improve the model performance, and that errors in CS points appear in the same proportion streaming and offline ST. Our work sets baseline results and provides insight into the impact of CS on the performance of different models, and helping to identify potential points for future research that can contribute to the advancement of the field. To sum up, the main contributions of our work are:

- We provide baseline results for streaming ST for CS speech, contrary to previous work that focuses on offline settings.

- We provide baseline results to CS ST into a third language, contrary to previous work that focuses on monolingual transcription. To do so, we extend the Fisher-Miami CS dataset, adding Spanish and German targets.

## 2 Related Work

During the past few years, there has been an increasing interest in CS tasks. Prior work has focused in MT (Sinha and Thakur, 2005; Winata et al., 2021; Zhang et al., 2021; Yang et al., 2020) and ASR (Lyu et al., 2006; Ahmed and Tan, 2012; Vu et al., 2012; Johnson et al., 2017; Yue et al., 2019). However, the topic of CS in ST has been relatively under-explored, and usually concentrating only on monolingual transcription (Nakayama et al., 2019; Hamed et al., 2022; Weller et al., 2022), and relying on synthetically generated data (Nakayama et al., 2019; Huber et al., 2022).

The first work on CS ST was done by Nakayama et al. (2019). The authors analyse different architectures and training configurations for Japanese-English CS to English monolingual transcription.

Weller et al. (2022) present a similar work but in a different language pair. The authors present a CS dataset with natural English-Spanish CS text and speech sources and English text targets, gathering CS sentences in Fisher and Bangor Miami datasets. With these data, they are able to evaluate ASR and

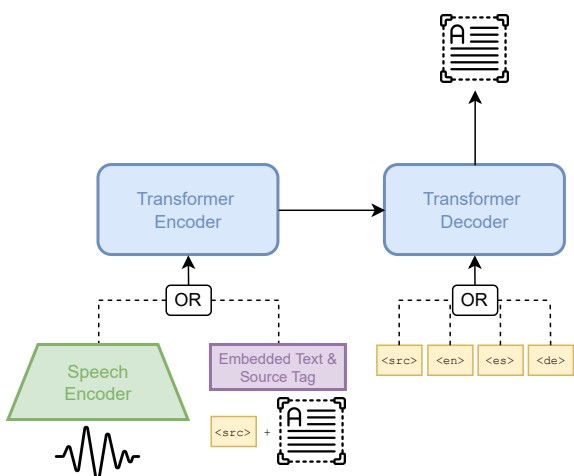

Figure 1: Proposed model architecture. The multimodal encoder supports training on both speech translation and text translation data. The tagging scheme is designed to allow generating either the (code-switched) transcript or a (monolingual) translation.

ST, although the ST setting is actually monolingual transcription. The authors explore different architectures through a two-steps training: a pretraining on non-CS data and a fine-tuning on CS data. They find that end-to-end ST models obtain higher accuracy than cascaded ones and that accuracy on CS test sets improves after the fine-tuning step without noticeably impacting performance on non-CS sets.

Later, Hamed et al. (2022) present a corpus for Egyptian Arabic-English CS tasks. The dataset contains text and speech CS sources, and targets in monolingual English and Egyptian Arabic. By combining these sets the authors are able to study ASR (from CS speech to CS text), as well as MT and ST. However, because of the target languages, both the ST and MT settings are actually monolingual transcription and a text-to-text variant of this task.

Finally, Huber et al. (2022) present LAST, a language-agnostic model for ST and ASR that aims to replace acoustic language ID gated pipelines by a unique CS model. However, their work focuses on inter-sentential CS (when a CS happens just at sentence boundaries) using synthetic data.

## 3 Model

We adopt the multimodal model design proposed by Ye et al. (2021) for speech translation (Figure 1). This model supports speech transcription, speech translation, and text translation, and leverages paired data of all three tasks through multitask

training. Similar to Ye et al. (2021), we extract speech representations using a pretrained wav2vec 2.0 BASE model (Baevski et al., 2020)[2] which results in 20ms per frame. To compute downsampled speech representations, wav2vec 2.0 applies a stack of three convolutional layers, resulting in 160ms per frame: each layer has a kernel of 3 and a stride of 2. To extract text representations for multitask text-to-text training, we simply use a 1024-dimensional embedding layer. Next we attach an encoder-decoder Transformer (Vaswani et al., 2017) with pre-layer normalization, a hidden dimension of 1024, dropout of 0.1, five encoder layers and three decoder layers. The input to the encoder is either the downsampled speech representations, or the embedded source text. In the decoder, we use 1024-dimensional LSTMs (Hochreiter and Schmidhuber, 1997) instead of self-attention which obtained better results in preliminary investigations.

The model is trained in a multi-task fashion, where we sum the losses of the transcription task, text translation task, speech translation task, as well as a CTC loss (Graves et al., 2006) applied on top of the full encoder. Tasks are weighted equally.

Importantly to our work, we use a shared decoder to perform either transcription or translation, with a language tag indicating the desired output language for ST, or the tag <src> to generate a transcript. Note that the transcript will be equivalent to the translation in the source language for monolingual sentences, but a special token for transcripts is needed to account for CS sentences.

To employ our model in a streaming setting, we use the re-translation technique (Niehues et al., 2018; Weller et al., 2021). This technique re-translates the utterance to update its prior prediction as additional information is received. To control the trade-off between latency, flickering, and accuracy, we set a mask on the last $k$ sub-words of the prior prediction, allowing the model to rewrite only that part of the output. Therefore, a high $k$ allows the model to rewrite the whole prediction, obtaining a high accuracy but poor latency and flickering scores, and on the contrary, setting $k = 0$ forces the model to commit to the previous prediction, hindering the accuracy but leading to no flickering and the lowest possible latency. Section 5 contains experiments to obtain the appropriate $k$.

---

[2]Specifically, `facebook/wav2vec2-base-960h` via `Hugging Face Transformers` (Wolf et al., 2020).

## 4 Datasets

| Pre-training | | | |
|---|---|---|---|
| **Dataset** | **Language** | **Source** | **#Samples** |
| MuST-C | En-Es | Original | 270 000 |
| | En-De | Original | 234 000 |
| CoVoST | Es-En | Original | 64 351 |
| | De-En | Original | 71 831 |
| | En-De | Original | 232 958 |
| | Es-De | Synthetic | 64 351 |
| | De-Es | Synthetic | 71 831 |
| Fisher | Es-En | Original | 130 600 |
| Miami | Es-En | Original | 6 489 |

| Fine-tuning | | | |
|---|---|---|---|
| **Dataset** | **Language** | **Source** | **#Samples** |
| Fisher | En/Es-En | Original | 7 398 |
| | En/Es-Es | Synthetic | 7 398 |
| | En/Es-De | Synthetic | 7 398 |

Table 1: Summary of the training data used during our two-steps training.

Although our primary target is CS speech, we train our models on both monolingual and CS data due to the scarcity of the latter. In particular, we use the following datsets:

**Bangor Miami (Deuchar et al., 2014):** The dataset contains recorded conversations between bilingual English/Spanish speakers in casual settings, with a high proportion of naturally occurring code-switched speech. The recordings were obtained using small digital recorders worn on belts, resulting in low audio quality with background noise. We use the splits for CS ST defined by Weller et al. (2022).

**Fisher (Cieri et al., 2004):** The dataset was collected for ASR by pairing Spanish speakers located in the US and Canada through phone calls. Although it is not a CS focused dataset, it contains a significant amount of CS utterances due to the speakers being in English-speaking contexts. The recording was done through phone recordings in 2004, which makes it a noisy ASR dataset, although less noisy than Miami. We use the splits for CS ST defined by Weller et al. (2022).

**CoVoST (Wang et al., 2020):** A multilingual and diversified ST datset based on the Common Voice project (Ardila et al., 2020). This dataset includes language pairs from multiple languages into English, and it includes low resource languages.

**MuST-C (Di Gangi et al., 2019):** A dataset for ST research. It is a large-scale, multi-language dataset that includes speech recordings from English TED Talks and corresponding human transcriptions and translations. The dataset covers translation from English to many languages. The recording context (TED talks) makes it a quality clean dataset.

### 4.1 Data Collection

Miami and Fisher CS sets consist of a source in CS En/Es, along with CS transcripts and monolingual English transcripts as targets. To expand the range of languages included, we include the monolingual Spanish transcript, as well as a new language not used in the source, namely German. By including this new language, we will be able to assess the performance of our models in pure speech translation, as opposed to previous work on monolingual transcription. Hence, we collect data for Miami and Fisher CS test and validation sets in German and Spanish. The data was translated by professional translators who were native speakers in the respective target languages.

### 4.2 Data Usage and Preparation

Following (Weller et al., 2022), we divide our experiments in two steps: (1) pre-training on monolingual data and, (2) fine-tuning on code switched data.

During the pretraining we use CoVoST (Es-En, De-En, En-De splits), MuST-C (En-Es, En-De splits) and the non-CS sets in Fisher and Miami datasets (Es-En). Additionally, we use MarianMT [3] model from Hugging Face Transformers package (Wolf et al., 2020) to translate CoVoST De-En set to Spanish, and Es-En set to German, obtaining data for the pairs Es-De and De-Es. During the finetuning step, we focus on Fisher's code-switched (Es/En-En) training set (7389 samples) and extend it for Es/En-Es and Es/En-De translation using the MarianMT model to translate English targets to German and Spanish.

We use 200 epochs for the pretraining stage and 100 epochs for finetuning. We use the Adam

---

[3]We manually clean the translations afterward.

---

(Kingma and Ba, 2015) optimizer with $\alpha = 5e{-}4$, $\beta_1{=}0.9$, $\beta_2{=}0.98$. For pretraining, we use an inverted square root learning schedule with 500 warm-up steps. For finetuning a tri-stage schedule with 12.5% warm-up steps, 12.5% hold steps, and 75% decay steps.

For the experiments with prefix sampling, we use the same training set but prefix-sampling half of the instances following the approach presented by Niehues et al. (2018). For a summary of the data used on each step see Table 1.

## 5 Experiments

Our experiments follow four main directions: (1) Finding a reasonable $k$ to control re-translation flickering and latency, (2) studying the occurrence of errors around CS switching points, (3) analyzing the usefulness of prefix-sampling and (4) establishing baseline numbers for translation to a third language and for streaming tasks for CS speech, including transcription, monolingual translation, and translation.

To evaluate our models we will use three different metrics. To measure the model accuracy we use BLEU (Papineni et al., 2002) with SACREBLEU (Post, 2018) and a beam size of 5. To evaluate the lag between model input and output we use Average Lag (AL, Ma et al. (2019)), and to measure the flickering we use Normalized Erasure (NE, Arivazhagan et al. (2020)). Additionally, we use WER to evaluate ASR performance.

### 5.1 Metrics Trade-off and $k$ Analysis

As described in Section 3, our model uses re-translation (Niehues et al., 2018) to generate a streaming output. Following the re-translation approach, we mask the last $k$ sub-words of an output when predicting the following one. We evaluate latency, flickering and accuracy metrics for $k \in \{0, 5, 10, 15, 20, 25, 30, +\infty\}$. As shown in Figure 2, results are consistent for Fisher and Miami datasets and across the different language pairs. All metrics increase together with $k$. However the gap between 30 and $+\infty$ is much higher in AL and NE than in BLEU. BLEU shows improvements for higher $k$ but it is more stable than the other metrics. For this reason, we henceforth use $k = 15$, since BLEU scores are close to optimal while NE and AL are still low.

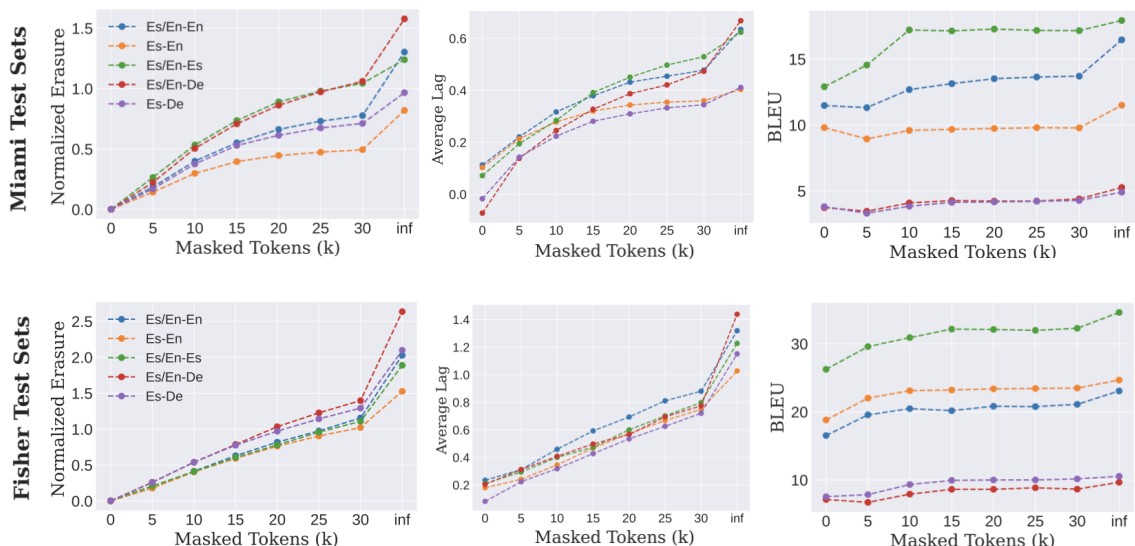

Figure 2: BLEU, Normalized Erasure and Average Lag scores under different streaming constraints. In each prediction step, the model has to commit to the previous prediction except for the last $k$ tokens (sub-words). We evaluate the performance of the model for $k \in \{0, 5, 10, 15, 20, 25, 30, +\infty\}$.

| | | Fisher | | | | | Miami | | | | |
| | | | CS | | Mono. | | | CS | | Mono. | |
| | Model | En | Es | De | En | De | En | Es | De | En | De |
|---|---|---|---|---|---|---|---|---|---|---|---|
| BLEU(↑) | FISHER CS | 23.3 | 30.3 | 12.2 | 22.9 | 12.8 | 19.7 | 16.0 | 6.4 | 11.9 | 5.9 |
| | FISHER CS W/ PREFIXES | 23.7 | 30.9 | 12.2 | 22.0 | 13.0 | 22.1 | 18.3 | 7.0 | 13.9 | 6.7 |
| | (WELLER ET AL., 2022) † | 25.6 | - | - | 26.1 | - | 14.7 | - | - | 17.6 | - |
| AL(↓) | FISHER CS | 0.6 | 0.5 | 0.6 | 0.5 | 0.5 | 0.5 | 0.4 | 0.4 | 0.4 | 0.3 |
| | FISHER CS W/ PREFIXES | 0.5 | 0.5 | 0.5 | 0.5 | 0.5 | 0.5 | 0.5 | 0.5 | 0.4 | 0.4 |
| NE(↓) | FISHER CS | 1.2 | 1.2 | 1.3 | 1.1 | 1.4 | 1.2 | 1.2 | 1.4 | 1.0 | 1.2 |
| | FISHER CS W/ PREFIXES | 1.2 | 1.0 | 1.2 | 1.2 | 1.0 | 1.0 | 1.0 | 1.1 | 1.6 | 0.8 |

Table 2: BLEU, Average Lag (seconds), and Normalized Erasure scores in streaming **Speech Translation**, for trainings with and without prefix sampling. In every experiment we set $k = 15$. †: Best results reported by Weller et al. (2022) in offline ST.

## 5.2 Code-Switches and Errors in Predictions

We hypothesize that CS points are points of high linguistic uncertainty and, therefore, comparably hard to predict or translate. Hence, words around CS switch points would tend to be predicted wrong. We analyze this phenomenon for an ASR task comparing offline and streaming models with the aim of: (1) confirming or denying that more wrong predictions happen near CS points, (2) studying how offline or streaming ST can affect the conclusion of (1).

We analyze the predicted transcripts of our model in the ASR [4] task on Fisher CS test set under three different inference constraints: a streaming model with $k = 0$ (which has no flickering and the

[4]Note that this can only be evaluated in ASR (not ST), because of the need of a CS target.

lowest possible latency), a streaming model with $k = 15$ (which we have found to be a reasonable choice to obtain a better accuracy without a critical effect on flickering and latency) and an offline model (which would be equivalent to a streaming model where $k = +\infty$). We establish a recall-based metric and count words in the reference transcript as predicted right if the word appears in the predicted transcript, and as predicted wrong otherwise. We study the proportion of words that are predicted right and their distance (in words) to a CS point. Hence, those words at a distance of 1 are right before or after a CS, and so on. To do so, we define the $Recall$ at distance $d$ as:

$$R(d) = \frac{right\_pred(d)}{right\_pred(d) + wrong\_pred(d)} \quad (1)$$

| | Model | Fisher | | | | | Miami | | | | |
|---|---|---|---|---|---|---|---|---|---|---|---|
| | | | CS | | Mono. | | | CS | | Mono. | |
| | | En | Es | De | En | De | En | Es | De | En | De |
| BLEU(↑) | FISHER CS | 41.8 | 45.8 | 24.27 | 35.5 | 23.7 | 49.4 | 41.8 | 19.9 | 31.7 | 19.5 |
| | FISHER CS W/ PREFIXES | 41.8 | 44.1 | 22.9 | 35.7 | 22.5 | 48.1 | 38.7 | 19.1 | 32.2 | 18.9 |
| AL(↓) | FISHER CS | 0.4 | 0.4 | 0.4 | 0.4 | 0.4 | 0.2 | 0.2 | 0.2 | 0.2 | 0.2 |
| | FISHER CS W/ PREFIXES | 0.4 | 0.4 | 0.4 | 0.4 | 0.4 | 0.2 | 0.2 | 0.2 | 0.2 | 0.2 |
| NE(↓) | FISHER CS | 0.06 | 0.04 | 0.06 | 0.04 | 0.04 | 0.00 | 0.00 | 0.00 | 0.00 | 0.00 |
| | FISHER CS W/ PREFIXES | 0.04 | 0.04 | 0.06 | 0.04 | 0.04 | 0.00 | 0.00 | 0.00 | 0.00 | 0.00 |

Table 3: BLEU, Average Lag (seconds), and Normalized Erasure scores in streaming **Text Translation**, for trainings with and without prefix sampling. In every experiment we set $k = 15$.

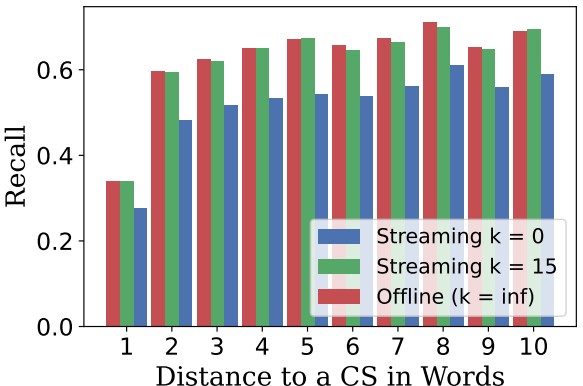

Figure 3: Analysis of errors in the prediction of words for different distances to a CS point under different inference constraints.

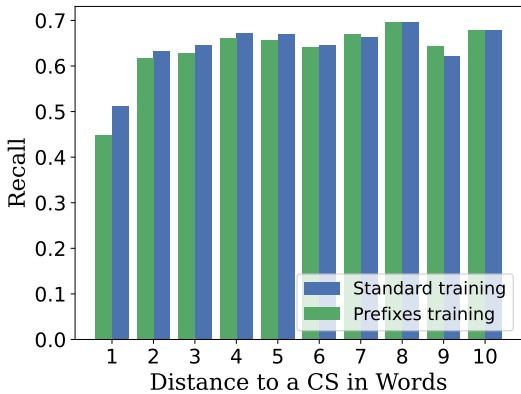

Figure 4: Analysis of errors in the prediction of words for different distances to a CS point, with and without prefix-sampling the training set.

The results in Figure 3 show that CS points impact the model's accuracy. Those words at a distance of 1 are predicted wrong in the highest proportion for every model. However, starting from $d = 2$, the recall increases only slightly, or stays close to constant, so the effect of a CS does not last long. Secondly, we also see that although the streaming setting with $k = 0$ has an overall worse recall, having less available context when making the predictions does not affect those words close to CS points more than those that are not. In particular, we see that the drop between $d = 2$ and $d = 1$ is lower for the streaming model with $k = 0$. This indicates that, contrary to what we expected, the lack of context in streaming ST does not have a negative impact on CS points, and therefore, the model needs the same context to properly predict CS or not CS words.

### 5.3 Usefulness of Prefix-sampling

A frequently used technique to train streaming models consists of sampling prefixes from part of the training data. We study the impact of using this technique in accuracy, latency, and flickering metrics and its impact on errors around CS points.

To analyze the usefulness of this training strategy, we compare a model trained on the Fisher CS set against a model trained on the same set but substituting half of the complete utterances by prefixes. As shown in Table 2, prefix-sampling produced an improvement in BLEU scores, especially in Miami test sets (up to +2.4). Surprisingly, this training strategy that aims to improve the performance in latency or flickering worsens the Average Lag scores and does not significantly impact Normalized Erasure.

Furthermore, we study whether prefix sampling impacts the accuracy of the predictions around CS points. In Figure 4, we use the same *recall* metric as in Section 5.2 to compare both models. We see that prefix training degrades the accuracy of the predictions around CS points, especially in those words at a distance of 1, where the recall drops from 0.51 in the standard training to 0.45 in prefixes training.

|  | | Fisher | | Miami | |
|---|---|---|---|---|---|
|  | Model | CS | Mono | CS | Mono |
| WER(↓) | FISHER CS | 34.9 | 29.8 | 63.3 | 63.5 |
|  | FISHER CS W/ PREFIXES | 35.4 | 29.9 | 60.6 | 58.1 |
| AL(↓) | FISHER CS | 1.0 | 0.8 | 0.8 | 0.6 |
|  | FISHER CS W/ PREFIXES | 0.5 | 0.4 | 0.5 | 0.3 |
| NE(↓) | FISHER CS | 1.2 | 1.0 | 1.2 | 1.1 |
|  | FISHER CS W/ PREFIXES | 1.1 | 0.8 | 1.2 | 0.6 |

Table 4: WER, Average Lag (seconds), and Normalized Erasure scores in streaming **Automatic Speech Recognition**, for trainings with and without prefix sampling. In every experiment we set $k = 15$.

## 5.4 Performance Analysis

After the experiments described in previous sections, we have found that using prefix-sampling does not lead to a noticeable performance improvement. Furthermore, we have seen that masking the last 15 sub-words in each step during the translation of a sentence shows an optimal trade-off between the different evaluation metrics. Since there is no previous work in CS streaming ST, we can not fairly compare our results to previous work, and therefore we aim to set baseline numbers. However we compare the BLEU scores of our model to the scores obtained by (Weller et al., 2022) for offline ST to English (Table 2), to analyse if the performance drop between offline and streaming ST is reasonable. As expected, our streaming model suffers a performance degradation in most of the test sets compared to the offline model in previous work. However, CS ST to English in the Miami dataset obtains an improvement of up to +7.4 BLEU.

When analyzing the performance of German translation we see that there is an important drop compared to English and Spanish translation (both present on the source). CS Speech Translation is commonly studied and evaluated just in translation to languages present in the source, therefore we believe that the performance drop in German is a relevant finding that shows the importance of not relying just on monolingual transcription when aiming for CS ST and sets a baseline result for further work in translation to a third language. Regarding Average Lag and Normalized Erasure, we present our results as a baseline, since previous work using Fisher and Miami datasets was done in offline tasks. However, to have an estimation of the quality of our model in these metrics, we compare our scores with the ones obtained by Weller et al. (2021) on MuST-C data, which are over 1 for both metrics. In Table 2, we can see that we obtain similar scores, therefore we conclude that the performance of our model is reasonable regarding flickering and lag.

## 5.5 Results in Machine Translation and Automatic Speech Recognition

Although the main scope of this work in Speech Translation, we evaluate our models for Machine Translation and Automatic Speech Recognition too. We can easily do this given that the model we are using is multitask and allows us to work on each of the three settings by switching the the input type and properly defining the a tag to generate the output.

In Table 3 we can see the results obtained for MT. We see that, as in ST, prefix sampling does not improve AL and NE scores. Furthermore, in the case of MT using prefixes degrades the performance of the majority of the models. Regarding BLEU scores, we observe that as in ST those tasks that consist on translating to a language present in the source obtain a much higher accuracy than those where we translate to German.

In Table 4 we see the results for the ASR setting. In this case, prefix sampling does work as expected regarding AL and NE scores, being the models with prefixes the ones with lower scores. However, it still has a negative impact on the performance of the models, specially in Miami test sets. Regarding WER, the scores obtained for the Miami dataset are much worse than the ones obtained by Fisher ones, a pattern that we have not observed in translation tasks. This could be due to the fact that during the pretraining, the data used for translation tasks comes from many different datasets, allowing the model to properly learn to generalize. However, the available data with CS targets corresponds mostly to the Fisher dataset (130 600 samples), compared to only 6 487 from the Miami dataset (see Table 1 for more details on the data distribution).

## 6 Conclusions

In this work, we have tackled two open ends in CS ST: translation to a third language and streaming settigns. To do so, we have trained offline and streaming models for direct translation and transcription of CS speech. Furthermore, we have extended Fisher and Miami test and validation sets with new Spanish and German targets. By doing this we have been able to analyse not only monolingual transcription, but also pure translation. We have observed a drop of up to 18 BLEU points between the two settings, showcasing the importance of not relying on monolingual transcription when aiming for ST models, as has been commonly done in previous work. Given the greater complexity of translating to a third language as compared to monolingual translation, we think that incorporating additional data would be necessary to tackle the accuracy drop. However, since natural code-switched data is limited and generating synthetic data is beyond the scope of this study, we leave this for future research.

To summarize, our work presents new data, an in depth analysis of the impact of CS in the predictions, and results for streaming CS Speech Translation and translation to a third language, which can serve as a baseline for future work in a field that although relevant is still far from solved.

## Limitations

Our work is limited to high-resource languages such as English, German, and Spanish. Therefore, further work needs to be done tackling low resource languages in order to achieve real-world CS translation.

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
