# OpenReview forum: "Towards Real-World Streaming Speech Translation for  Code-Switched Speech"
_EMNLP/2023/Workshop/CALCS — EMNLP 2023 Workshop CALCS_

### Official Review · Reviewer_e3HX · 2023-10-02
**Well written paper with interesting baseline contribution to Code Switching in Speech translation**

**Rating:** 4
**Confidence:** 3

**Review:**

**Summary**

This paper explores the problem of code switching in speech translation. The author's primary contributions are as follows:
1. Provides baseline results for code switching in speech translation to a 3rd language (language different from the ones used in input)
1. Provides baseline results for code switching in speech translation in a streaming setting and compare specified metrics with offline speech translation.
1. Extends existing code switched speech translation datasets to include annotations for a third language, to enable evaluation for the 1st contribution mentioned above.

Overall, the paper is very well written and the authors set very valuable baselines for CS through extensive experimentation. They do not propose new model design, but rather use the one discussed in [Ye et al (2021)](https://www.isca-speech.org/archive/interspeech_2021/ye21_interspeech.html). The paper is well related to the objectives of the workshop as it contributes new CS resources and shares baseline numbers for future research.

Pros:
- The paper is very well written and contains all details needed by others to reproduce this research with some effort.
- The experiments for streaming speech translation are very interesting. I really enjoyed the section on metrics tradeoff and the results of k-analysis. This was a good introduction for readers who do not have enough background on streaming speech translation.
- The results on distance to CS words are interesting as they clearly highlight the problem immediately in the vicinity of CS words. They also show that these problems apply to both streaming and offline settings.

Cons:
- The authors mention using LSTMs in the decoder of their model instead of self-attention. They should specify more details on how better they found this to be in their preliminary investigation. It would also be interesting to look up if others working with speech translation or MT have had similar observations.
- The authors consistently find that prefix-sampling degrades performance on the recall metrics as well as on MT. The authors do not discuss any intuitions for why this is happening. This makes me question the purpose of adding the prefix sampling section to the paper?

Misc:
- Typo on line 180. `some` -> `sum`

**Candidate For Best Paper:**

No

**Reason For Best Paper:**

The paper tackles 2 unstudied aspects of code switching in speech translation: speech translation to a third language and the streaming setting.

**Related:**

5: It is very related to the workshop.

---

### Official Review · Reviewer_cqZS · 2023-10-03
**Author's address the real-time speech translation on CS corpus with different languages, but additional information about created datasets and additional evaluation is necessary.**

**Rating:** 3
**Confidence:** 5

**Review:**

The work described in this passage addresses Online speech translation with two key challenges. They are:

Translation to a Third Language: The first challenge involves translating  CS speech into a third language (Spanish and German). The authors have developed both offline and streaming models to tackle this challenge.

Streaming Settings: The second challenge pertains to handling streaming settings for CS speech. Streaming translation and transcription are more demanding due to real-time requirements, and the authors have worked on models to address this issue.

Pros:

The authors addressed the importance of streaming settings when translating into monolingual/CS text and it was interesting aspect to research in a CS language.

Cons

Limited New Target Dataset Information: The passage in line 259 mentions that, "The data was translated by professional translators who were native speakers in the respective target languages", but no information is added about inter annotator aggrement or any other metrics to prove that their generated data is "gold standard".

To enhance the quality and comprehensibility of the paper, it is imperative to provide a detailed explanation of why the models performed better on Spanish datasets compared to German target datasets. This analysis would shed light on the factors influencing the system's performance and contribute to a more insightful discussion.

Additionally, since the authors have conducted tests on text translation, it is crucial to include specific examples that illustrate the system's output under two contrasting scenarios: a scenario where the translation is deemed good and another where it is considered suboptimal. This would allow readers to directly assess the quality and limitations of the system.

For instance, the paper could include a sample source input, along with the corresponding target output in both Spanish and German. In the case of a "good translation," the authors should explain what elements or factors contributed to the accuracy and fluency of the output. Conversely, for the "suboptimal translation," a thorough analysis should be provided to elucidate the shortcomings or errors made by the model.

By offering concrete examples and detailed explanations for the performance variations, the paper can provide a clearer understanding of the strengths and weaknesses of the system, thereby enhancing the overall quality of the research.

**Candidate For Best Paper:**

No

**Reason For Best Paper:**

N/A

**Related:**

5: It is very related to the workshop.

---

### Official Review · Reviewer_otVK · 2023-10-03
**The paper introduces an approach and resource for studying code-switching speech translation in a streaming setting. It explores a scenario where the target language is not one of the source languages in the CS data. The paper is well-written and includes multiple experiments and detailed analyses of the results. However, there are specific points of concerns which I highlight below.**

**Rating:** 4
**Confidence:** 4

**Review:**

The paper presents an approach and a resource to study CS speech translation in streaming setting. The authors also introduced a setting where the target language is not one of the source languages in the CS data.
The paper is overall well-written and presents several experiments and interesting analyses of results.
I have the following comments:

- one of the main contributions of the paper is the introduction of data in another target language. While the authors stated that existing data (validation and test) was translated by professional translators, do they ensure any quality control? How many instances are translated in total and by how many translators?

- The paper lack ablation experiments: On the MTL setings, do all tasks impact performance measures positively?  What is the effect of using additional translated data for German and Spanish?

**copy editing errors**
- line 134 "higher accuracy that" -> "...than"
- line 138: "through through" -> "through"
- line 180: "some of losses" -> "sum of losses"
- line 233/245: remove "quality" or consider rephrasing
- line 296: "lattency" -> "latency"

**Candidate For Best Paper:**

No

**Reason For Best Paper:**

Not a candidate for best paper

**Related:**

5: It is very related to the workshop.